# Redirect Tropism of Fowl Adenovirus 4 Vector by Modifying Fiber2 with Variable Domain of Heavy-Chain Antibody

**DOI:** 10.3390/genes15040467

**Published:** 2024-04-08

**Authors:** Yongjin Wang, Xiaohui Zou, Xiaojuan Guo, Zhichao Zhang, Min Wang, Tao Hung, Zhuozhuang Lu

**Affiliations:** 1NHC Key Laboratory of Medical Virology and Viral Diseases, National Institute for Viral Disease Control and Prevention, Chinese Center for Disease Control and Prevention, Beijing 100052, China; 2School of Public Health, Baotou Medical College, Inner Mongolia University of Science and Technology, Baotou 014040, China

**Keywords:** adenoviral vector, flow adenovirus 4, VHH, fiber, tropism, simian adenovirus 1, transduction, foldon

## Abstract

The variable domain of a heavy-chain antibody (VHH) has the potential to be used to redirect the cell tropism of adenoviral vectors. Here, we attempted to establish platforms to simplify the screening of VHHs for their specific targeting function when being incorporated into the fiber of adenovirus. Both fowl adenovirus 4 (FAdV-4) and simian adenovirus 1 (SAdV-1) have two types of fiber, one of which is dispensable for virus propagation and is a proper site for VHH display. An intermediate plasmid, pMD-FAV4Fs, was constructed as the start plasmid for FAdV-4 fiber2 modification. Foldon from phage T4 fibritin, a trigger for trimerization, was employed to bridge the tail/shaft domain of fiber2 and VHHs against human CD16A, a key membrane marker of natural killer (NK) cells. Through one step of restriction-assembly, the modified fiber2 was transferred to the adenoviral plasmid, which was linearized and transfected to packaging cells. Five FAdV-4 viruses carrying the GFP gene were finally rescued and amplified, with three VHHs being displayed. One recombinant virus, FAdV4FC21-EG, could hardly transduce human 293 or Jurkat cells. In contrast, when it was used at a multiplicity of infection of 1000 viral particles per cell, the transduction efficiency reached 51% or 34% for 293 or Jurkat cells expressing exogenous CD16A. Such a strategy of fiber modification was transplanted to the SAdV-1 vector to construct SAdV1FC28H-EG, which moderately transduced primary human NK cells while the parental virus transduced none. Collectively, we reformed the strategy of integrating VHH to fiber and established novel platforms for screening VHHs to construct adenoviral vectors with a specific tropism.

## 1. Introduction

Adenoviruses are non-enveloped viruses containing a genome of linear, double-stranded DNA of 26–48 kb in length. Adenoviridae is classified into six genera, among which Mastadenovirus and Aviadenovirus infect mammalian hosts and birds, respectively [1]. Adenoviral vectors have been widely used in biological research, gene therapy, and vaccine development [2,3,4,5]. A cell tropism is one of the key characteristics to be considered with priority in adenoviral vector utilization.

The knowledge of adenovirology accumulates mainly through the study of human adenovirus C (HAdV-C2 or -C5). The icosahedral virion of adenovirus contains 240 capsomeres of hexon trimers on the 20 triangular facets and 12 pentameric penton capsomeres at each vertex [6,7]. For Mastadenovirus, 12 fibers protrude from the pentons, each a trimer of the fiber polypeptide. In contrast, a penton of Aviadenovirus contains two identical or different fiber trimers [8,9]. Fibers are the major ligand used by adenovirus to bind its cellular receptor for entry into the host cell. Fiber–receptor binding is the initial step for adenovirus to infect the host cell, which makes fiber the most important player for the cell tropism of adenovirus [10,11]. Mature fiber is a homotrimer and is composed of domains of the tail, shaft, and knob. The tail is located at the N-terminal of fiber to be embedded into the penton base. The knob is the distal terminal domain being used to interact with the cellular receptor, while the shaft connects the tail and knob [12,13]. Therefore, the modification of the fiber knob will lead to the change in the cell tropism of adenovirus.

The HAdV-5 vector is the most commonly used adenoviral vector due to proven technologies in vector construction, packaging, and preparation. Tropism modification is often carried out on an HAdV-5 vector system, and the fiber is the major target for modification [14,15]. The immediate idea is fiber replacement. While HAdV-5 employs the coxsackie virus and adenovirus receptor (CAR) on the cell membrane as an initial receptor [16,17], HAdV-B fibers bind to CD46 or DSG2 [18]. For example, replacement of the fiber knob with that from HAdV-B35 will assign the HAdV-5 vector a capability to transduce cells expressing CD46. However, the amount of adenovirus receptors is limited, which restricts the scope and effect of utilizing this approach. Oligopeptides were inserted into the HI loop of the HAdV-5 fiber knob, and such modification would direct HAdV-5 to cells expressing the target molecule the oligopeptide binds [19,20]. Usually, this method will not abolish the original tropism of HAdV-5. In addition, the integration of a peptide into the fiber knob might make the peptide lose its binding ability due to steric hindrance and thus lead to an unexpected outcome. Techniques have also been developed to fuse a peptide to other molecules on the virion surface, such as hexon, penton base and pIX protein [20]. However, no approach can guarantee an adenoviral vector with a desirable tropism so far.

Antibodies are generally considered as a group of the best biomolecules for detecting and targeting purposes due to their characteristics of high affinity and high specificity. Camelid animals, such as llamas, alpacas, and camels, evolve to have the ability to produce a group of immunoglobulins called heavy-chain antibodies (HCAb), the molecule of which contains only two heavy chains and completely lacks the light chain [21,22]. The variable domain of the heavy chain of HCAb (VHH) determines the specific recognition and binding to an antigen. VHH can be cloned and produced as a single peptide chain with complete function by using a prokaryotic or eukaryotic protein expression system. VHH, commonly named a single-domain antibody (SdAb) or nanobody, possesses many advantages over a traditional antibody or single-chain variable fragment (scFv), including small size, low immunogenicity, thermal resistance, unfolding reversibility, proteolytic resistance, and high solubility [21]. Procedures have been presented to fuse VHH to the fiber protein of an HAdV-5 vector for modifying the tropism [23,24].

We have established vector systems based on fowl adenovirus 4 (FAdV-4) and simian adenovirus 1 (SAdV-1) to avoid the disadvantage of pre-existing antibodies against HAdV-5 in human beings [25,26,27]. FAdV-4, one type from Aviadenovirus, is different from HAdV-5 in that FAdV-4 protrudes two fiber trimers encoded by diverse fiber genes from one penton capsomere while HAdV-5 possesses only one fiber trimer in each penton capsomere [8,28,29]. SAdV-1, a rare serotype from Mastadenovirus, also has two fiber genes [30,31]. Therefore, SAdV-1 has two types of penton capsomeres, from which different fiber trimers extend. Natural killer (NK) cells are characterized as cytotoxic lymphocytes of the innate immune system, and they have now become a group of important target cells in cellular immunotherapies because of the extensive cell sources and the off-the-shelf settings. On the other hand, it is also well known that NK cells are extremely resistant to the transduction of an exogenous gene [32]. CD16A is a key membrane marker of NK cells, and some VHHs against CD16A have been screened and sequenced [33]. Here, we attempted to establish platforms based on two-fiber adenoviruses for selecting appropriate VHHs to redirect the tropism of an adenoviral vector. Three VHHs against CD16A were chosen as the tentative molecules for fiber modification and used in this study [33].

## 2. Materials and Methods

### 2.1. Cells, Viruses, Plasmids, and Oligonucleotides

For the cultivation of LMH cells, flasks or plates were pre-treated with 0.1% gelatin (Cat. G9391, Sigma-Aldrich, St. Louis, MO, USA) in water at 37 °C for half an hour according to the instructions of American Type Culture Collection (ATCC). Chicken LMH cells (Leghorn Male Hepatoma; ATCC CRL-2117), human 293 (ATCC CRL-1573), and 293SE13 cells were cultivated in Dulbecco’s modified Eagle’s medium (DMEM) plus 10% fetal bovine serum (FBS; HyClone, Logan, UT, USA) at 37 °C in a humidified atmosphere supplemented with 5% CO_2_. DMEM plus 2% FBS was used as a maintenance medium for virus rescue or amplification. 293SE13 cells are a derivative strain of 293 cells, which constitutively expressed SAdV-1 E1B55K protein and served as the packaging cells for replication-defective SAdV-1 viruses. Adherent cells were detached by trypsin treatment, 1:3 split, seeded in new flasks, and would be used for plasmid transfection or virus infection when they reached 70%-90% confluency the next day. Jurkat (T-cell leukemia) cells were similarly cultivated except that DMEM in the culture medium was changed to RPMI 1640. NK-92 cells (ATCC CRL-2407) were cultivated in α-MEM plus 0.2 mM myo-Inositol (Cat. 17508, Sigma), 0.02 mM folic acid (Cat. F8758, Sigma), 12.5% heat-inactivated horse serum (Cat. 26050070, Gibco, Carlsbad, CA, USA), 12.5% heat-inactivate FBS (10100-139, Gibco), and 200 U/mL rhIL-2 (Cat. 130-097-746, Mitenyl Biotec, Gaithersburg, MD, USA). Cells were regularly split twice a week.

FAdV4-CG was a recombinant FAdV-4 prepared previously in the laboratory, which carried deletions of ORF1, ORF1b, and ORF2 in the genome and an insertion of a CMV-promoter-controlled GFP (CG) expression cassette [25]. pKFAV4S-GFP, an adenovirus plasmid, carries FAdV-4 genomic DNA with deletions of ORF0, ORF1, ORF1B, ORF2, and ORF19A; and at the left deletion site in the genome, a CMVp-controlled GFP expression cassette was added, which was flanked with SwaI dual sites to facilitate promoter or transgene replacement [34]. pMD-FAV4Fs is an intermediate plasmid previously constructed for FAdV-4 fiber modification, which contains a major part of fiber1 and complete fiber2 coding sequences (CDS) of FAdV-4 [29]. SAdV1-EG was an E1/E3-deleted SAdV-1 vector, which carried a human EF1a-promoter-controlled GFP expression cassette in the original E1 region. Intermediate plasmid pKSAV1H5E4p-AscI, which contains fiber genes of SAdV-1, and adenoviral plasmid pKSAV1H5E4p-EG were constructed previously [27]. Overlap extension PCR was routinely performed for site-directed mutation in small plasmids, and traditional PCR or TaqMan real-time PCR were performed for an identification or detection aim. Single-stranded DNA oligos were synthesized and used as the primers in PCR. The primer sequences and related information for PCR are summarized in Table 1.

### 2.2. Preparation of Primary NK Cells

Diluted anticoagulant-treated blood from a healthy donor was layered on the Ficoll solution, and peripheral blood mononuclear cells (PBMCs) were collected from the buffy coats after low-speed centrifugation. PBMCs were washed twice with NK MACS Medium (Cat. 130-114-429, Mitenyl Biotec); suspended in NK MACS Medium supplemented with 5% of heat-inactivated autologous plasma, 1000 IU/mL rhIL-2, and 1000 IU/mL rhIL-15 (Cat. 130-95-766, Mitenyl Biotec) at a density of 1.5 × 10^6^ cell/mL; and cultivated in flasks coated by an anti-human CD16 monoclonal antibody. Two volumes of a fresh growth medium were added the next day, and the cells were cultivated for another 8 days without extra operation. After that, the cells were diluted with a growth medium to achieve a density of 5 × 10^5^ cells/mL and routinely split every 2 days to maintain the ideal cell density [35,36]. The expression of CD3, CD56, and CD16A was monitored, and the percentage of CD3-CD56+ cells was often more than 90%. After 2 or 3 weeks of cultivation, the amplified NK cells were ready for the transduction of adenoviral vectors.

### 2.3. Construction of Adenoviral Plasmids

The method of combined restriction digestion and Gibson assembly (restriction-assembly) was used to construct adenoviral plasmids [27]. The procedure for fiber modification was divided into two steps. First, the fiber2 gene in an intermediate plasmid was replaced with a synthesized DNA fragment or product of overlap extension PCR. Second, the modified fiber genes in the intermediate plasmid were used to replace original genes in the adenoviral plasmid by restriction-assembly. For fiber modification in FAdV-4 vectors, a DNA fragment was chemically synthesized, which consisted of a sequence-encoding partial coiled-coil and foldon motif in fibritin protein of bacteriophage T4, sequence-encoding VHH of C21 against human CD16A, sequence-surrounding AvrII site, and sequence-surrounding HindIII site in the pMD-FAV4Fs plasmid. The fragment between the AvrII/HindIII site in pMD-FAV4Fs was replaced with the synthesized DNA by restriction-assembly to generate pMD-FAV4FS-CFC21. pMD-FAV4FS-CFC21 was digested with KpnI/EcoRV, pKFAV4S-GFP was digested with MauBI/SbfI, the products were mixed, and Gibson assembly was performed to generate pKFAV4CFC21-GFP (NEBuilder HiFi DNA Assembly Master Mix, Cat. E2621, New England Biolabs, Ipswich, MA, USA). Overlap extension PCR was carried out to generate a DNA fragment containing the human EF1a promoter (EF1ap) and GFP CDS, which was used to replace CMV-promoter-controlled GFP (CMVp-GFP) between two SwaI sites in pKFAV4CFC21-GFP by restriction-assembly to generate pKFAV4CFC21-EG. To further remove the coiled-coil motif in modified fiber2, the sequence between BamHI/NaeI was mutated by overlap extension PCR, and a modified plasmid of pMD-FAV4FS-FC21 was generated by restriction-assembly. Similarly, modified fiber2 could be transferred to pKFAV4CFC21-EG by restriction-assembly. The procedures of constructing other FAdV-4 adenoviral plasmids are illustrated in Appendix A. VHH of C28 against CD16A was selected to modify fiber2 in SAdV-1 to generate pKSAV1FC28H-EG adenoviral plasmid (Appendix A), and a 6× His tag was fused to the C-terminal of modified fiber2 to facilitate the detection of the fiber protein by Western blot.

### 2.4. Virus Rescue, Purification, Titration, and Identification

FAdV-4 adenoviral plasmids, such as pKFAV4CFC21-EG or pKFAV4FC21-EG, were digested with PmeI, recovered, and used to transfect LMH cells. Foci formed by GFP-positive cells would be observed under a fluorescence microscope 3 to 5 days post transfection. The rescued viruses were further amplified in LMH cells. The viruses released to the culture medium or associated with the cells were harvested separately and subjected to density gradient ultracentrifugation on an iodixanol medium. The purification procedure was described in detail elsewhere [26]. pKSAV1FC28H-EG was linearized by SwaI and used to transfect 293SE13 cells. The rescued SAdV1FC28H-EG virus was amplified in 293SE13 cells and purified with traditional CsCl density gradient ultracentrifugation [37].

The purified viruses were lysed in a buffer containing 10 mM EDTA, 0.1% SDS, and 0.2 mg/mL proteinase K (pH 7.6) at 50 °C for 2 h. The concentration of virus genomic DNA was determined with the Qubit double-stranded DNA assay kit (Cat. Q32851, Thermo Fisher Scientific, Waltham, MA, USA). The mass concentration was converted to the molar concentration of the virus genome, and finally the concentration of viral genome copies (the viral particle titer) was calculated [26]. The multiplicity of infection (MOI) was calculated from particle titers in this study. Infectious titers (IU/mL) of FAdV-4 viruses or SAdV1FC28H-EG were determined with the limiting dilution assay on LMH or 293 cells, respectively.

Virus genomic DNA was recovered from lysed purified virus (Genomic DNA Clean & Concentrator kit, Cat. D4010; Zymo Research, Irvine, CA, USA) and digested with restriction enzymes. The products were resolved on 0.7% agarose gel containing ethidium bromide by electrophoresis, and the DNA fragments were visualized on a UV transilluminator and photographed as digital digestion maps by using a CCD camera. The modification regions and DNA assembly sites were amplified by PCR and further confirmed by sequencing the PCR products.

### 2.5. Western Blot

The expression of modified fiber of F2-FC28H on FAdV4FC28H-EG virions was evaluated by Western blot. Purified viruses of 1 × 10^10^ viral particles (vp) were mixed with 2× SDS gel-loading buffer (100 mM Tris-Cl, 4% sodium dodecyl sulphate, 0.2% bromophenol blue, 20% glycerol, pH 6.8) with or without 200 mM dithiothreitol, incubated in a boiling water bath for 5 min or unheated, resolved on precast SDS-PAGE gel (4–20%, Cat. P0057A, Beyotime Biotechnology, Beijing, China) by electrophoresis, and transferred to nitrocellulose membranes. A mouse monoclonal antibody against 6× His (Cat. TA-02, ZSGB-BIO, Beijing, China) was 1:2000 diluted in PBST (10 mM PBS, 0.05% Tween 20) containing 5% skimmed milk and used as the primary antibody, and HRP-conjugated goat anti-mouse IgG (ZB-2305, ZSGB-BIO) was 1:5000 diluted and used as the secondary antibody. The bands were developed by covering the membrane with Amersham ECL Prime Western blotting Detection Reagent (Cat. RPN2232, GE, Buckinghamshire, UK) and photographed in a dark cabinet by using a CCD camera (Tanon, Shanghai, China).

### 2.6. Plaque-Forming Assay

Recombinant FAdV-4 viruses of 5000 or 10,000 vp were diluted in 1 mL maintenance medium and added to LMH cells seeded in a 6-well plate. After two hours of incubation, the virus diluents were removed, the cells were washed twice with DMEM, and 2.5 mL semisolid culture media of DMEM plus 2% FBS and 1% low-melting agarose were added to each well. A fresh maintenance medium of 2 mL per well was supplemented to replenish nutrients 4 days post infection. The foci formed by GFP-positive cells were photographed under a fluorescence microscope 5 days post infection. The areas of all foci in each well were measured by using the Fiji image processing tools (http://fiji.sc/, accessed on 4 April 2024) [38]. The median sizes of the foci formed by different viruses were compared by using the Mann–Whitney nonparametric test.

### 2.7. Preparation of Lentivirus Vector

The coding sequence of human CD16A (GenBank NM_000569) was chemically synthesized and used to replace the fragment of Tet-on3G/IRES/Neo in pLVX-EF1a-Tet3G (Clontech Laboratories, Mountain View, CA, USA) to generate plasmid pLVX-EF1a-CD16A (Appendix A). Plasmids of pLVX-EF1a-CD16A, psPAX2, and pMD2.G were mixed to transfect 293T cells by using a transfection reagent of jetPRIME (Polyplus, Illkirch, France) according to the manufacturer’s instructions. The transfection medium was removed and fresh DMEM plus 2% FBS and 7 mM sodium butyrate was added 5 h post transfection. After another 40 h of cultivation, culture supernatant containing LVX-EF1a-CD16A lentivirus vector was collected, centrifugated at 5000× *g* for 5 min to remove cellular debris, aliquoted, and frozen at −80 °C. The infection titer was determined by quantifying the integrated viral genome with TaqMan real-time PCR (Table 1) in lentivirus-infected 293 cells [39].

### 2.8. Lentivirus Vector Transduction and Immunofluorescence Assay

Lentivirus LVX-EF1a-CD16A was diluted in a fresh culture medium containing 8 µg/mL polybrene (Cat. H9268, Sigma) and used to transduce 293, Jurkat, or NK-92 cells. A fresh culture medium of 2 volumes was supplemented 12 h post infection. The culture medium containing the lentiviral vector was replaced with a fresh culture medium 24 h post infection. The expression of CD16A was monitored by flow cytometry assay. Briefly, the adherent cells were detached by trypsin treatment, suspended in a blocking solution of PBS containing 1% BSA for half an hour, collected after centrifugation at 500× *g* for 5 min, re-suspended in a 1:100 diluted PE-conjugated primary antibody against human CD16A (mouse monoclonal antibody, Cat. 10389-MM41, Sino Biological, Beijing, China) in PBS plus 1% BSA with stirring on a laboratory rocker for 20 min at 4 °C in the dark, washed with PBS plus 1% BSA, and subjected to a flow cytometry assay. In parallel, PE-conjugated mouse IgG1 (Cat. 12-4714-42, Invitrogen, Waltham, MA, USA) was used to replace the anti-CD16A antibody in sample preparation and served as an isotype control.

### 2.9. Gene Transduction Assay

Adherent cells were seeded in plates the day before transduction, while exponentially growing suspension cells were collected by centrifugation, counted, and directly used for transduction experiments. Viruses were diluted in a maintenance medium and used to infect cells in wells of a 24-well plate at a volume of 0.25 mL/well. The plate was placed on a shaker at 37 °C and rocked for 4 h at a frequency of 10 times per minute. For adherent cells, the virus diluent was aspirated and a fresh maintenance medium of 0.5 mL was added to each well. Forty-eight hours post infection (calculated from the start of virus incubation), cells expressing GFP were observed under a fluorescence microscope, and the cells were detached by trypsin treatment (for adherent cells) or harvested by centrifugation (for suspension cells), suspended in 10 mM phosphate-buffered saline (PBS) plus 1% FBS and 1.5% paraformaldehyde, and temporarily reserved at 4 °C. The fixed cells were subjected to a flow cytometry assay within one week.

### 2.10. Statistical Analysis

The data are presented as the mean ± SD and analyzed with a one-way or two-way analysis of variance unless otherwise indicated. *p* < 0.05 was considered statistically significant.

## 3. Results

### 3.1. Construction of Recombinant Adenoviruses with VHH Displayed on the C-Terminal of Fiber

The knob domain has been postulated to be essential for the trimerization of the fiber [40], while the foldon domain (30 aa) of bacteriophage T4 fibritin is known for the function to trigger trimerization [41]. David T. Curiel’s group used the C-terminal of fibritin (95 aa), including the last two coiled-coil segments and foldon, to bridge the fiber shaft of HAdV-5 fiber and scFV or VHH [42,43]. We followed their strategy; and at the same time, we also tried to use foldon without the coiled-coil region. As shown in Figure 1A, the knob sequence of FAdV-4 fiber2 was firstly replaced with synthesized DNA (CFC21) encoding the coiled coil/foldon and C21 VHH against human CD16A in the intermediate plasmid. Next, the modified fiber2 was transferred to the adenoviral plasmid by restriction-assembly. Finally, the promoter of CMV was changed to that of human EF1a to control the expression of the GFP reporter gene. The same strategy was repeated to change the coiled coil/foldon to foldon, C21-VHH to C28H-VHH, and so on (Figure 1A,B and Appendix A). The amino acid sequence of fiber2 modified by foldon and C21 VHH is shown in Figure 1C. To validate that knob/VHH replacement could be transplanted from Aviadenovirus to Mastadenovirus, the fiber2 in the SAdV-1 adenoviral plasmid was modified with foldon/C28H (Figure 1B and Appendix A).

After transfecting packaging cells with linearized adenoviral plasmids, recombinant viruses were successfully rescued. The rescued viruses spread on the monolayer of LMH or 293SE13 cells, leading to the formation of GFP foci and cytopathic effects (CPEs) at the end (Figure 2A and Appendix A). After amplification and purification, virus genomic DNA was extracted from purified viral particles and identified by restriction analysis. For the sake of convenience, short virus names are used for VHH-incorporated FAdV-4 viruses in the present study (Figure 1B). The restriction maps of CFC21-EG and FC21-EG are shown in Figure 2B,C, respectively, and those of other recombinant viruses are shown in Appendix A. The incorporation of VHH-modified fiber2 in virions was representatively inspected on FC28H-EG, which carried a 6× His tag at the distal C-terminal of modified fiber2 and could be conveniently detected by a Western blot. Modified fiber2 of FC28H-EG formed a trimer, which could not be depolymerized by SDS at ambient temperature (Figure 2C).

### 3.2. Propagation of Recombinant Adenoviruses in Packaging Cells

FAdV-4 possesses two different fiber trimers in each penton capsomere. Fiber1 is essential for virus propagation while fiber2 is dispensable [29]. It is unknown if the modification of fiber2 with VHH would affect the growth of recombinant viruses. After amplification and purification, the yield of virus harvested from six 15 cm dishes was calculated and is shown in Table 2. The yield for each recombinant FAdV-4 virus was more than 1 × 10^12^ vp, and the yields were at the same order of magnitude for all FAdV-4 viruses. Notably, the yield only resulted from a single, unoptimized experimental process for each virus and thus still had room for improvement. Plaque forming ability was further evaluated, and the median sizes of plaques formed by VHH-incorporated FAdV-4 viruses were similar to that formed by FAdV4-CG. Some viruses, such as CFC21-EG and FC21-EG, even formed relatively larger plaques (Figure 3). Collectively, the modification of fiber2 with VHH did not remarkably affect the propagation ability of FAdV-4 viruses in LMH cells.

### 3.3. Transduction of 293 Cells or CD16A-Expressing 293 Cells

293 cells were infected with a lentivirus vector carrying the human CD16A gene (LVX-EF1a-CD16A), and the transiently transduced cells were designated as 293CD16A. Flow cytometry data demonstrated that 98.8% of 293CD16A expressed CD16A 3 days post infection while there were no CD16A molecules on the membrane of parental 293 cells (Figure 4A). The percentage of CD16A-positive cells was sustained at a level of more than 90% for 293CD16A in the following 3 weeks, and during this period, adenovirus transduction experiments were performed (Figure 4B and Appendix A). It was seen that no recombinant FAdV-4 could effectively transduce 293 cells. The highest transduction efficiencies were found in FAdV4FC28-EG-infected cells, which were 2.1% or 21.0% for MOIs of 1000 or 10,000 vp/cell, respectively. All VHH-modified FAdV-4 could transduce 293CD16A more effectively except FC13-EG. For example, at the MOI of 1000 vp/cell, CFC21-EG and FC21-EG transduced 57.8% and 51.2% 293CD16A cells, respectively. The efficiencies rose to 94% at the MOI of 10,000 vp/cell. The parental virus of FAdV4-CG could transduce neither of the cells. These data indicated that modifying fiber2 with VHHs of C21 or C28 gave FAdV-4 the ability to transduce CD16A-expressing 293 cells specifically.

### 3.4. Transduction of CD16A-Expressing Suspension Cells

CD16A is a membrane marker of NK cells, which are a group of suspension leukocytes. Therefore, we further evaluate the transduction of VHH-modified FAdV-4 in Jurkat, a cell line of acute T-cell leukemia, and NK-92, a natural killer cell line.

Original Jurkat cells express no CD16A, while CD16A was detected in 97.5% lentivirus LVX-EF1a-CD16A-infected Jurkat-CD16A cells 3 days post infection (Figure 5A). Recombinant FAdV-4 viruses could hardly transduce Jurkat cells. Even when the MOI value increased to 10,000 vp/cell, the transduction efficiency remained as low as 1%, the background level. In contrast, in Jurkat-CD16A cells, the highest transduction efficiency was found for CFC21-EG, with values of 8.9%, 39.4%, and 67.0% at MOIs of 100, 1000, and 10,000 vp/cell, respectively (Figure 5B–D and Appendix A). The results from FC21-EG were very close to that from CFC21-EG, followed by that from FC28H-EG, FC28-EG, FC13-EG, and control FAdV4-CG. In NK-92 or CD16A-expressing NK92-CD16A cells, the transduction trends were similar to that in Jurkat or Jurkat-CD16A although the efficiency reduced overall. The highest transduction efficiency was approximately 13% when CFC21-EG or CFC21-EG were used at an MOI of 10,000 vp/cell (Figure 6A–C). These data demonstrated that, to some extent, the incorporation of VHH against CD16A facilitated the transduction of FAdV-4 to CD16A-expressing suspension cells.

### 3.5. Transduction of Primary NK Cells

CD16A is one of the membrane markers of NK cells, and the results of flow cytometry confirmed the expression of CD16A in cultured primary NK cells (Figure 7A). We examined the transduction of VHH-modified FAdV-4 in NK cells. Unexpectedly, only background fluorescence of GFP was detected for all tested FAdV-4 (Figure 7B), indicating that VHH-modified FAdV-4 vectors failed to transduce or express an exogenous gene in primary NK cells.

### 3.6. Transduction Ability of VHH-Modified SAdV-1 Vector

SAdV-1 belongs to HAdV-G and possesses two types of fiber. We replaced the knob domain of fiber2 with foldon/VHH-C28 in E1/E3-deleted SAdV-1 to generate recombinant SAdV1FC28H-EG. The trimerization of modified fiber2 was identified by a Western blot (Appendix A). SAdV1FC28H-EG could transduce Jurkat-CD16A with higher efficiency than Jurkat cells. The percentages of GFP-positive cells were 8.1% or 26.4% for Jurkat or Jurkat-CD16A cells, respectively, when SAdV1FC28H-EG was used at an MOI of 1000 vp/cell (Figure 8A). When SAdV1FC28H-EG was used to transduce primary NK cells, it was seen that the transduction efficiencies were 4.8% or 15.1% at MOIs of 1000 or 10,000 vp/cell. The control virus of SAdV1-EG, which possessed original fibers, could hardly transduce primary NK cells, and the transduction efficiency was lower than 1% at either MOI (Figure 8B). The results indicated that incorporating VHH against CD16A enhanced the transduction of SAdV-1 to NK cells.

## 4. Discussion

For the modification of an adenovirus tropism, the first thing that comes to mind is fiber replacement [14]. However, the resource of adenovirus fibers is limited, and the number of cellular receptors of adenovirus is even smaller. The incorporation of a short peptide into capsid proteins, such as fiber, hexon, and pIX, is another option [20]. Such operation can attribute new targeting characteristics to an adenoviral vector without abolishing the native tropism. Therefore, it cannot satisfy the requirement of specificity in gene therapy where a native tropism will cause an adverse effect. Interaction between an antigen and antibody is of high specificity as well as high affinity. There is another advantage of using an antibody to recognize and bind to antigen protein: for any antigen, scientists are able to find specific monoclonal antibodies and clone the genes encoding them by following standard procedures. The immunoglobulin G (IgG) of mammals is a polymeric protein with a molecular weight of 150 kD, which is composed of two heavy and two light chains. It is unlikely to integrate intact IgG to adenoviral fiber. David T. Curiel and colleagues chose to fuse ScFv to the C-terminal of HAdV-5 fiber [44]. They achieved partial success. The cytoplasmic production process of ScFv is not compatible with the assembly of adenovirus in the nucleus, which led to low yield of functional virions. Using VHH instead of ScFv ameliorated this flaw. However, the difficulty in preparing recombinant HAdV-5 was not completely solved, because VHH replacement made HAdV-5 lose the ability to infect and propagate in 293 cells. A derivative of 293 cells that expressed HAdV-5 fiber was established and used for the first several rounds of virus amplification [24,42]. It made the virus amplification complicated and tedious, which could be the reason why the fiber/VHH strategy did not become popular.

A high prevalence of pre-existing immunity in humans hampers the use of HAdV-5 vectors in gene therapy or vaccine development [45,46]. We established adenoviral vectors based on FAdV-4 and SAdV-1, against which human beings have no pre-existing immunity [25,27]. FAdV-4 and SAdV-1 possess another unique property that distinguishes them from most other adenoviruses: they express two types of fiber on the virion [8,31]. This property allows the virus with deletion in one fiber gene to be able to infect and propagate in the packaging cells, thus facilitating the modification of another fiber gene [29,30]. FAdV-4 fiber1 is the essential fiber gene for virus growth in fowl cells, and in contrast, it is completely impotent to mediate the transduction of FAdV-4 to human cells [26]. Therefore, FAdV-4 is an ideal model to evaluate the function of VHH-integrated fiber.

NK cells are important target cells for cellular immunotherapy while CD16A is one of the key membrane markers of NK cells [47]. In addition, some sequences of single-domain antibodies against CD16A have been published [33]. Therefore, we evaluated the possibility to construct NK-cell-targeting FAdV-4 by incorporating VHHs against CD16A into fiber2.

The plan was carried out in three steps. First, in order to trigger the trimerization of modified fiber effectively, we compared the strategy of using foldon alone and the existing strategy of using a coiled coil/foldon to bridge the tail/shaft of FAdV-4 and VHH. Both viruses (CFC21-EG and FC21-EG) were successfully rescued, there was no significant difference in virus growth between them, and the specific transduction to CD16A-expressing 293 cells was improved. Since the foldon domain (30 aa) is shorter than a coiled coil/foldon (95 aa), the strategy of using foldon alone was employed for the following experiments. Second, we constructed more FAdV-4 viruses to integrate different VHHs (C28 and C13), which were reported together with C21 by the same research group. Besides the goal of screening more effective VHHs, we attempted to transplant such fiber modification to SAdV-1, a serotype from the genus of Mastadenovirus. All recombinant viruses were rescued, they propagated to a considerable titer in the packaging cell lines, and the trimerization of VHH-integrated fibers was confirmed.

Finally, we assessed the specific transduction of these viruses to CD16A-expressing cells. All FAdV-4-derived viruses could hardly transduce tested human cells, including 293, Jurkat, NK-92, and primary NK cells; and some of the viruses, such as CFC21-EG and FC21-EG, transduced CD16A-expressing 293 or Jurkat (293CD16A or Jurkat-CD16A) with an efficiency of around 50% at a moderate MOI of 1000 vp/cell. However, CFC21-EG and FC21-EG were only able to transduce 13% of CD16A-expressing NK-92 cells at an MOI of 10,000 vp/cell, and other viruses were even inferior. None of these VHH-modified FAdV-4 viruses had the ability to transduce primary NK cells, which expressed endogenous CD16A. These results were unexpected. We then tested SAdV-1 with VHH-C28-modified fiber2 (SAdV1FC28H-EG) on primary NK cells. The situation improved, and the transduction efficiencies reached 5% or 15% when SAdV1FC28H-EG was used at MOIs of 1000 or 10,000 vp/cell, respectively. The parental SAdV1-EG could hardly transduce NK cells. These data clearly demonstrated that VHH incorporation attributed very high transduction specificity to adenoviral vectors. However, the transduction efficiency was relatively low and could not meet the requirement for basic biomedical research and cellular immunotherapy.

Adenovirus entry into cells experiences several steps, including virus attachment, endocytosis, escape from endosomes, cytoplasmic transport, and nuclear import [10,48,49,50]. Moreover, cells have evolved innate immunities to resist the infection of adenovirus. Failure in either step will lead to the abortive expression of the exogenous gene carried by the adenoviral vector. Although VHH incorporation helped the adenoviral vector attach to the target cell specifically, co-receptors or various intracellular environments might be responsible for the different outcome of the expression of the carried gene. Besides fiber–receptor interaction, SAdV-1 might evolutionally accumulate more advantages over FAdV-4 in transducing mammalian cells. For example, SAdV-1 possesses an RGD motif in its penton base, which interacts with the co-receptor of integrin on the host cell and helps virus infection [51,52]. In contrast, FAdV-4 does not have an RGD motif in the penton base. The FAdV-4 platform is useful for VHH screening, while the SAdV-1 vector system will benefit the function validation of selected VHH in tropism modification.

In conclusion, platforms based on two-fiber adenoviruses were established for evaluating the specific targeting of VHH-incorporated adenoviral vectors, and the use of a short peptide of foldon to trigger the trimerization of modified fiber was validated. Our work will benefit the screening of a single-domain antibody for constructing an adenoviral vector with a specific tropism.

## Figures and Tables

**Figure 1 genes-15-00467-f001:**
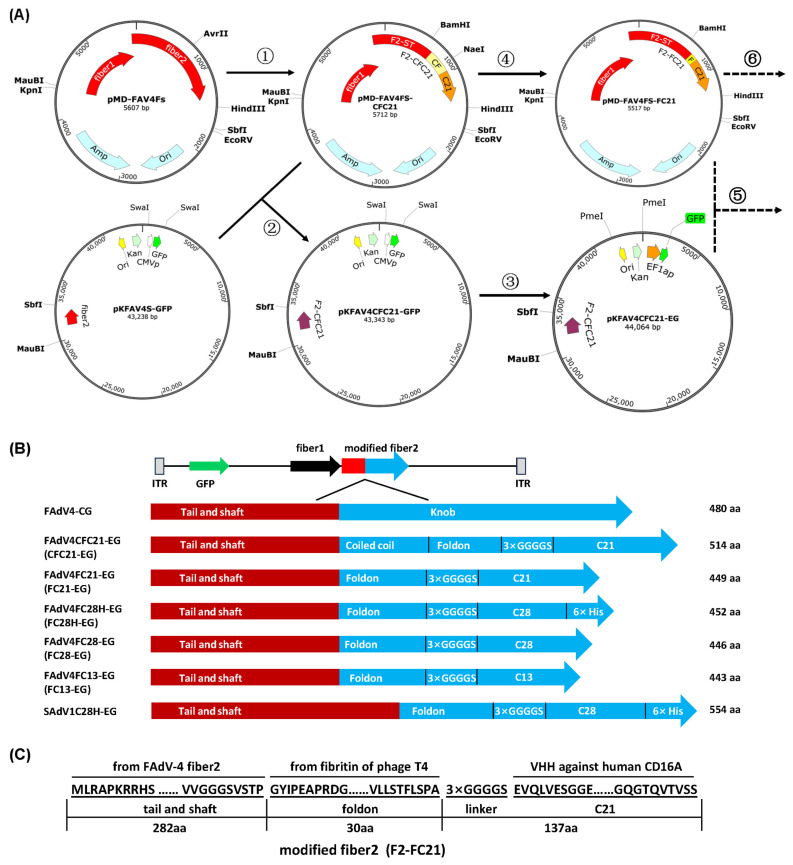
Schematic diagram of fiber modification for adenoviral vectors. (**A**) The procedure for the modification of the FAdV-4 fiber2 knob. The intermediate plasmid pMD-FAV4Fs and adenoviral plasmid pKFAV4S-GFP have been constructed previously. The detailed steps are described in the Section of Materials and Methods. (**B**) The diagram of the domains of the modified fiber 2 proteins. (**C**) Representative amino acid sequences of modified fiber2 protein from FAdV4FC21-EG. C13, C21, and C28: VHHs against human CD16A; CMVp: CMV promoter; EF1ap: human EF1a promoter; Coiled coil: the 12th-13th coiled-coil segments from fibritin of phage T4; F2-ST: FAdV-4 fiber2 shaft and tail; ITR: inverted terminal repeat.

**Figure 2 genes-15-00467-f002:**
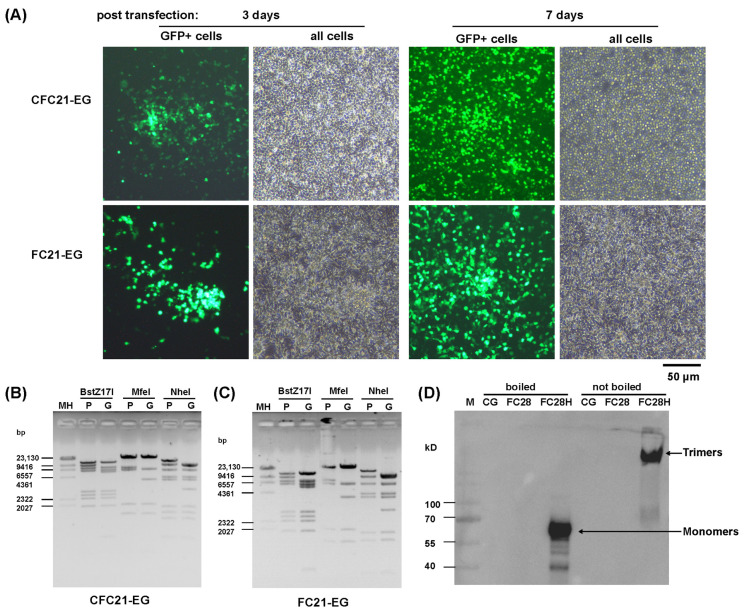
The rescue and identification of fiber-modified recombinant FAdV-4 viruses. (**A**) The rescue of VHH-C21-incorporated CFC21-EG and FC21-EG. PmeI-linearized adenoviral plasmids were used to transfect chicken LMH cells, and foci formed by GFP-positive cells were observed under a fluorescence microscope 3 or 7 days post transfection. The restriction analysis of genomic DNA extracted from purified CFC21-EG (**B**) or FC21-EG (**C**). Virus genomic DNA was digested with restriction enzymes and resolved on 0.7% agarose gel by electrophoresis. The corresponding adenoviral plasmid served as the control. The predicted molecular weights (bp) of digested fragments of the pKFAV4CFC21-EG plasmid were 12,141, 9046, 7166, 6118, 2755, 2545, 2337, and 1956 for BstZ17I; 27,094, 7000, 6130, 1983, 1630, and 227 bp for MfeI; and 14,005, 8952, 5858, 4143, 3805, 1885, 1659, 1463, and 973 for NheI. The predicted molecular weights (bp) of digested fragments of the CFC21-EG genome were 12,141, 7166, 6118, 5557, 2755, 2545, 2337, 1956, and 1018 for BstZ17I; 27,094, 6130, 3824, 1983, and 1630 for MfeI; and 8952, 8710, 5858, 4143, 3805, 2824, 1885, 1659, 1463, and 973 for NheI. (**C**) The predicted molecular weights (bp) of digested fragments of the pKFAV4FC21-EG plasmid were 12,141, 9046, 6971, 6118, 2755, 2545, 2337, and 1956 for BstZ17I; 27,094, 7000, 5935, 1983, and 1630 for MfeI; and 14,005, 8952, 5858, 4143, 3805, 1885, 1659, 1463, and 973 for NheI. The predicted molecular weights (bp) of digested fragments of the FAdV4FC21EG genome were 12,141, 6971, 6118, 5557, 2755, 25,45, 2337, 1956, and 1018 for BstZ17I; 27,094, 5935, 3824, 1983, and 1630 for MfeI; and 9978, 8952, 5858, 4143, 3805, 2824, 1885, 1659, and 973 for NheI. Molecular weights of fragments less than 800 bp are not given. MH: lambda/HindIII DNA marker. (**D**) The detection of VHH-integrated fiber by a Western blot. Equal amounts (1 × 10^10^ vp) of purified FAdV-4 viruses were mixed with a 2× SDS loading buffer, incubated in boiling water or not, loaded in SDS-PAGE gel, and transferred to nitrocellulose membranes after electrophoresis. The fiber protein of FC28H-EG was detected by a Western blot with an anti-6× His monoclonal antibody. CG: FAdV4-CG, FC28: FC28-EG, FC28H: FC28H-EG, M: protein marker.

**Figure 3 genes-15-00467-f003:**
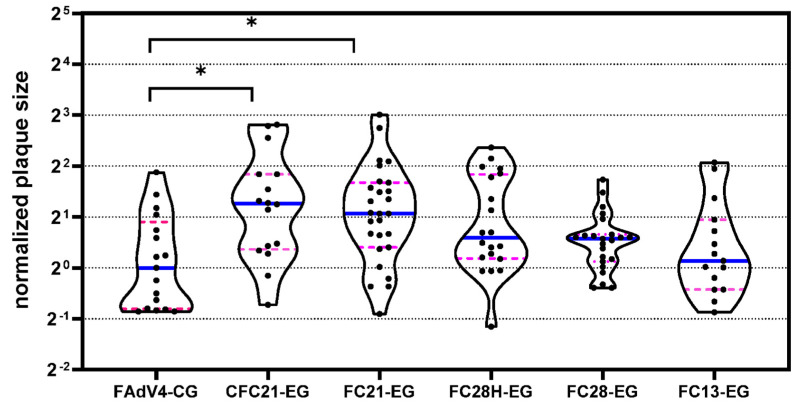
Violin plots of the plaque size data. FAdV4-CG was included in each batch of experiments, and all data collected from the same batch were normalized to the median size of plaques formed by FAdV4-CG. Normalized data were analyzed with the nonparametric Kruskal–Wallis test, and mean ranks were compared with that of FAdV4-CG (* *p* < 0.05).

**Figure 4 genes-15-00467-f004:**
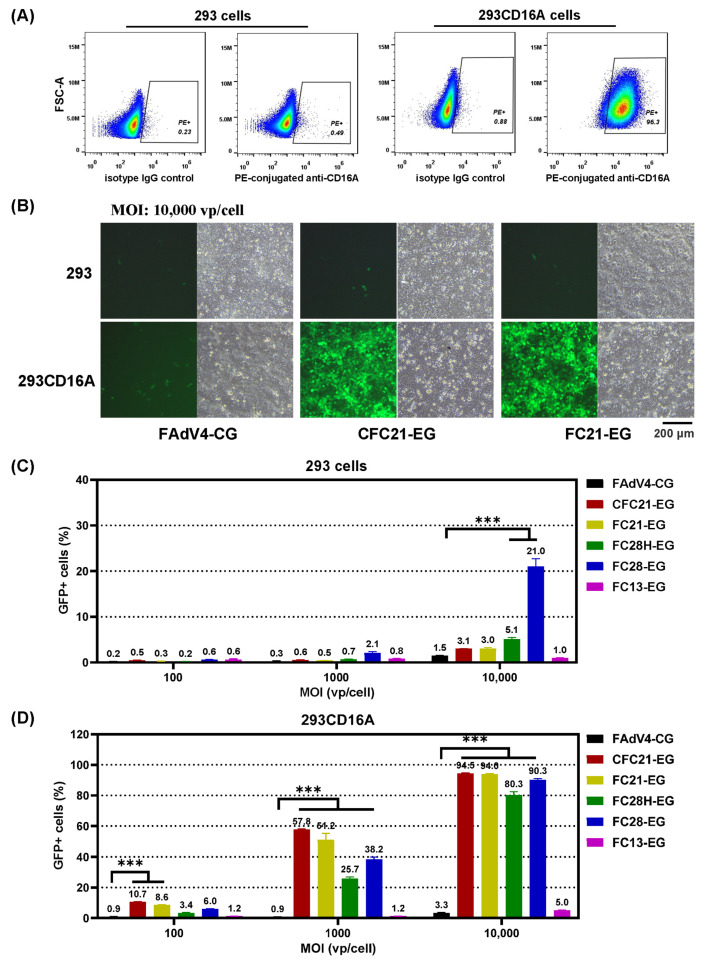
The transduction of CD16A-expressing 293 cells with fiber-modified FAdV-4 viruses. 293 cells were infected with a lentivirus vector carrying the exogenous human CD16A gene, and the expression of CD16A on the membrane was determined by flow cytometry with a PE-conjugated anti-CD16A monoclonal antibody 2 days post infection (**A**). Human 293 or 293CD16A cells were then transduced with VHH-incorporated FAdV-4 viruses at various MOIs for 4 h. GFP expression was firstly observed under a fluorescence microscope and photographed 48 h post infection. Some images are shown (**B**). The cells were detached and harvested immediately after the observation. The expression of GFP was determined by flow cytometry in virus-transduced 293 cells (**C**) or 293CD16A cells (**D**) (*** *p* < 0.001).

**Figure 5 genes-15-00467-f005:**
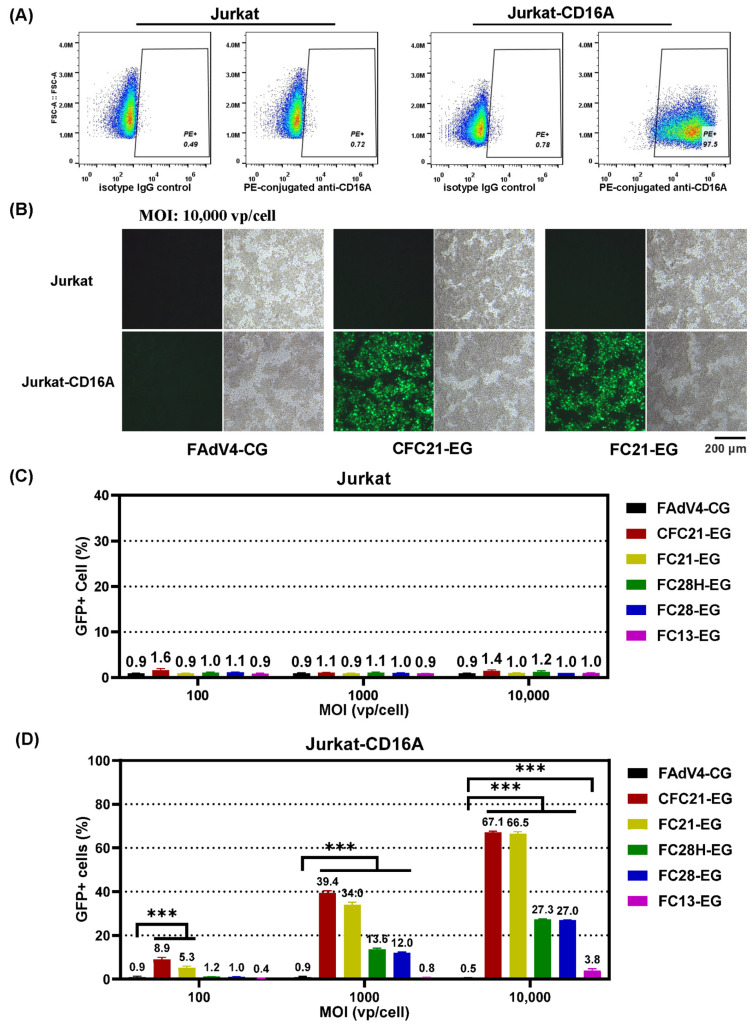
The transduction of suspension cell line Jurkat with fiber-modified FAdV-4 viruses. Human Jurkat cells were infected with a lentivirus vector carrying the EF1a-promoter-controlled CD16A gene, and the expression of CD16A was determined by flow cytometry with a PE-conjugated anti-CD16A antibody 48 h post infection (**A**). Parental Jurkat and CD16-expressing Jurkat-CD16A cells were then transduced with VHH-modified FAdV-4 vectors at various MOIs for 4 h. GFP expression was observed under a fluorescence microscope at 48 h post infection. Some images are shown (**B**). The cells were harvested after the observation and the expression of reporter GFP was analyzed by flow cytometry in FAdV-4-vector-transduced Jurkat cells (**C**) or Jurkat-CD16A cells (**D**) (*** *p* < 0.001).

**Figure 6 genes-15-00467-f006:**
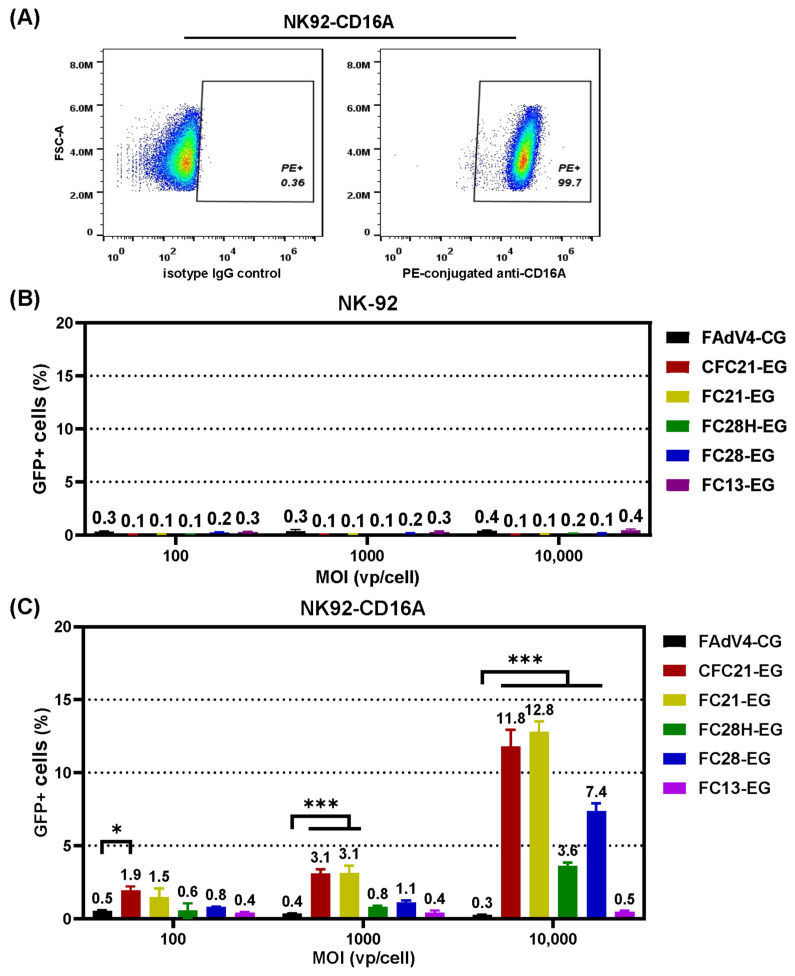
The transduction of human natural killer cell line NK-92 with fiber-modified FAdV-4 vectors. NK-92 cells were infected with a lentivirus vector carrying the human CD16A gene, and the expression of exogenous CD16A was confirmed by flow cytometry in infected cells (NK92-CD16A) 2 days post infection (**A**). NK-92 or NK92-CD16A cells were transduced by VHH-incorporated FAdV-4 viruses at various MOIs for 4 h. Cells were harvested 48 h post transduction, and the expression of GFP was assayed by flow cytometry in adenoviral-vector-transduced NK-92 cells (**B**) or NK92-CD16A cells (**C**) (* *p* < 0.05, *** *p* < 0.001).

**Figure 7 genes-15-00467-f007:**
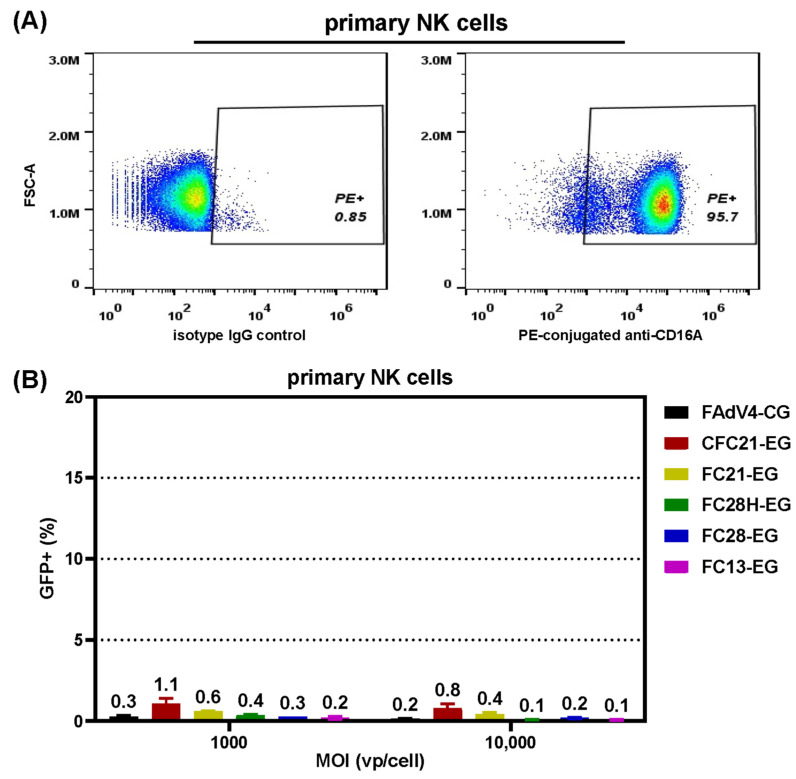
The transduction of primary NK cells with fiber-modified FAdV-4 vectors. Cultivated human primary NK cells were analyzed for the expression of CD16A on the membrane by flow cytometry (**A**). NK cells were transduced with VHH-modified FAdV-4 vectors at various MOIs for 4 h. The expression of the reporter GFP gene was firstly observed under a fluorescence microscope 2 or 3 days post transduction. The cells were harvested and the expression of GFP was assayed by flow cytometry 3 days post transduction (**B**).

**Figure 8 genes-15-00467-f008:**
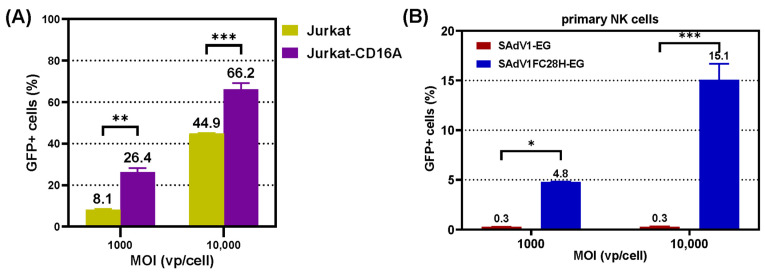
Transduction ability of the VHH-incorporated SAdV-1 vector. Jurkat or Jurkat-CD16A cells were transduced by SAdV1FC28H-EG at MOIs of 1000 or 10,000 vp/cell for 4 h, and the expression of GFP was assayed by flow cytometry 2 days post transduction (**A**). Similarly, primary NK cells were transduced with SAdV1FC28H-EG, and GFP expression was determined by flow cytometry. The parent vector of SAdV1-EG, which carried original fiber genes, served as a control (**B**) (* *p* < 0.05, ** *p* < 0.01, *** *p* < 0.001).

**Table 1 genes-15-00467-t001:** Summary information of PCR primers.

Fragment	Primer Name	Sequence	Template	Product (bp)	Annotation
EF1α-GFP	2207F2CF1KEGf	cgtcctttcg ttacagatct tcct	pKFAV4S-F2CF1K-EG	2126	
	2207F2CF1KEGr	cggtggatcg gatatcttat ctaga			
BamHI foldon	2304XCCfoldon-f	gtcaacaaca ctctggaagt gaaacc	NaeI-linearized pMD-FAV4FsF2cfC21	167	BamHI
	2304XCCfoldon-r	ggagcttcgg ggatgtagcc gggtgtggag acgctcccc			NaeI
FAV4F2FC28xH	2307F2FC28HxHf	ggccacggcc gtctactact gcaatg	pMD-FAV4FSF-C28H	110	EagI
	2307F2FC28HxHr	ggcctctaga ttatgaggag acggtgacct gggt			XbaI
SAV1F2C28Frag1	2307SAV1F2C28H1	gccattaact gtaagcaacg ggacccta	pKSAV1H5E4p-AscI	498	AvrII
	2307SAV1F2C28H2	gggatgtagc cgtcaggcga tgtaacttga ggagaa			
SAV1F2C28Frag2	2307SAV1F2C28H3	gttacatcgc ctgacggcta catccccgaa gctcct	pMD-FAV4FSF-C28H	558	XmnI
	2307SAV1F2C28H4	ggatgtcaaa tggttgtctt gaaacagttc ttagtggtga tgatgatggt gtgag			
WPREFrag	1309WPRE-F1	cctttccggg actttcgctt t	cell genomic DNA	176	
	1309WPRE-R1	gcagaatcca ggtggcaaca			
	1309WPRE-probe	actcatcgcc gcctgccttg cc			
RNasePFrag	1309RNaseP-F1	agatttggac ctgcgagcg	cell genomic DNA	65	
	1309RNaseP-R1	gagcggctgt ctccacaagt			
	1309RNaseP-probe	ttctgacctg aaggctctgc gcg			

**Table 2 genes-15-00467-t002:** Summary information of purified FAdV-4 and SAdV-1 vectors.

Virus Name	Short Name	Yield from Six 15 cm Dishes (×10^12^ vp)	Physical Titer (×10^11^ vp/mL)	Infectivity Titer (×10^9^ IU/mL)	Particle-to-IU Ratio
FAdV4-CG	-	5.3	35	8.4	420
FAdV4CFC21-EG	CFC21-EG	1.3	8.4	1.4	600
FAdV4FC21-EG	FC21-EG	2.4	16	8.5	190
FAdV4FC28H-EG	FC28H-EG	4.0	20	3.5	570
FAdV4FC28-EG	FC28-EG	1.1	10	3.0	330
FAdV4FC13-EG	FC13-EG	1.1	6.8	6.5	110
SAdV1-EG	-	0.28	5.1	19	30
SAdV1FC28H-EG	SFC28H-EG	0.52	5.8	1.5	390

vp: viral particle; IU: infectious unit.

## Data Availability

Data are contained within the article and Appendix A.

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
