# Peer review of "Redirect Tropism of Fowl Adenovirus 4 Vector by Modifying Fiber2 with Variable Domain of Heavy-Chain Antibody"

_genes, 2024, doi:10.3390/genes15040467_

Round 1

Reviewer 1 Report

Comments and Suggestions for Authors

In the manuscript “Redirect the tropism of fowl adenovirus 4 vector by modifying 2 fiber2 with variable domain of heavy-chain antibody” by Wang and collaborators, the authors explored the potential of using Variable Domain of Heavy-Chain Antibody (VHH) to modify tropism of adenoviral vectors (Fowl Adenovirus 4 (FAdV-4)). The authors incorporated VHHs against CD16A and were successful in human 293, Jurkat or NK-92  cells expressing CD16A (a receptor present in the surface of NK cells). However, the transduction was not successful in primary NK cells. Thereafter, the authors performed the same modification (incorporation of VHHs against CD16A) in Simian Adenovirus 1 (SAdV-1) and observed the transduction of up to 15% of primary NK cells.

The findings are somehow interesting, but the manuscript is mainly descriptive and lacks some in-depth discussion about the data. Below are some comments that may help to improve the manuscript:

1-) It is not clear the reason the authors decided to target receptors of NK cells. A rationale should be provided in the introduction.

2-) In the figure 2D, the authors showed the trimmers formation in samples that are not submitted to boiling process. However, they used SDS-PAGE gels and sample buffer with SDS. It will be important to demonstrate the formation of trimmers in non-denaturating conditions.

3-) The authors did not demonstrate the detection of VHH integrated fiber in the SAdV-1 (only FAdV-4 data is available). This data would be important to characterize this construction.

4-) It is not clear the reason why the authors jumped from FAdV-4 to SAdV-1. A rationale of this choice and discussion should be provided.

Reviewer 2 Report

Comments and Suggestions for Authors

In this manuscript, the authors describe a molecular strategy for redirecting the cell tropism of fowl adenovirus 4 by integrating the variable domain of the heavy chain of HCAb (VHH) in fiber2, which dictates cell receptor binding by adenovirus. This work allows for the design of novel platforms for screening VHHs for the construction of adenoviral vectors with specific and altered tropism. Overall, the experimental design of the study is sound and described in sufficient detail. My specific comments are provided below.

(1)    While the quality of the English language is acceptable, I would encourage the authors to have the manuscript edited/proofread to improve the quality and correct a number of grammatical errors.

(2)    Materials and Methods (line 134):  I assume the authors mean “healthy donor” for “health donor”; please confirm.

(3)    Materials and Methods (line 152):  The authors write, “modified fiber genes in the intermediate plasmid were brought back to adenoviral plasmid.” What the authors mean by “brought back” needs to be clarified or revised to improve clarity and specificity.

(4)    Results (line 295):  Clarify “rest recombinant viruses”.

Comments on the Quality of English Language

While the quality of the English language is acceptable, I would encourage the authors to have the manuscript edited/proofread to improve the quality and correct a number of grammatical errors.

Round 2

Reviewer 1 Report

Comments and Suggestions for Authors

The authors have satisfactorily answered the questions raised by this reviewer.